# Definitive Endodermal Cells Supply an in vitro Source of Mesenchymal Stem/Stromal Cells

Yumeng Zhang [1,2], Ye Yi[1,2], Xia Xiao[1,2], Lingling Hu[3], Jiaqi Xu[1,2], Dejin Zheng[1,2], Ho Cheng Koc[1,2], Un In Chan[3], Ya Meng[1,4], Ligong Lu [4], Weiwei Liu[1,5,6], Xiaoling Xu[3,6], Ningyi Shao [1,6], Edwin Chong Wing Cheung [3,6], Ren-He Xu [1,2,6✉] & Guokai Chen [1,2,6✉]

Mesenchymal stem/Stromal cells (MSCs) have great therapeutic potentials, and they have been isolated from various tissues and organs including definitive endoderm (DE) organs, such as the lung, liver and intestine. MSCs have been induced from human pluripotent stem cells (hPSCs) through multiple embryonic lineages, including the mesoderm, neural crest, and extraembryonic cells. However, it remains unclear whether hPSCs could give rise to MSCs in vitro through the endodermal lineage. Here, we report that hPSC-derived, SOX17+ definitive endoderm progenitors can further differentiate to cells expressing classic MSC markers, which we name definitive endoderm-derived MSCs (DE-MSCs). Single cell RNA sequencing demonstrates the stepwise emergence of DE-MSCs, while endoderm-specific gene expression can be elevated by signaling modulation. DE-MSCs display multipotency and immunomodulatory activity in vitro and possess therapeutic effects in a mouse ulcerative colitis model. This study reveals that, in addition to the other germ layers, the definitive endoderm can also contribute to MSCs and DE-MSCs could be a cell source for regenerative medicine.

[1] Centre of Reproduction, Development and Aging, Faculty of Health Sciences, University of Macau, Macau SAR, China. [2] Institute of Translational Medicine, Faculty of Health Sciences, University of Macau, Macau SAR, China. [3] Cancer Centre, Faculty of Health Sciences, University of Macau, Macau SAR, China. [4] Zhuhai Precision Medical Center, Zhuhai People's Hospital, Jinan University, Zhuhai, Guangdong, China. [5] Biological Imaging and Stem Cell Core Facility, Faculty of Health Sciences, University of Macau, Macau SAR, China. [6] MoE Frontiers Science Center for Precision Oncology, University of Macau, Macau SAR, China. ✉email: renhexu@um.edu.mo; guokaichen@um.edu.mo

Mesenchymal stem/stromal cells (MSCs) are multipotent stem cells that can be obtained from diverse human tissues and organs[1,2]. MSCs demonstrate great therapeutical values for various diseases, such as cardiovascular diseases[3–6], respiratory diseases[7,8], and gastrointestinal diseases[9,10]. MSCs from different organs display unique genetic features that are associated with their lineage origins[11–13]. MSCs can also be generated from human pluripotent stem cells (hPSCs), which emerges as a valuable source to produce MSCs for translational applications.

hPSCs include embryonic stem cells (hESCs) and induced pluripotent stem cells (hiPSCs), and they can theoretically differentiate into all cell types in the body[14]. hPSCs can also differentiate from MSCs in vitro through multiple lineages, such as the ectoderm[15], mesoderm[16], neuromesoderm[17], and extra-embryonic lineages[18]. However, there is no report on whether hPSCs could generate MSCs through the endoderm lineage. In contrast, MSCs have been successfully derived from endodermal organs, such as the lung, intestine, and liver[19,20]. Although it is generally believed that MSCs in endodermal organs come from the mesodermal origin, this concept has been challenged. For example, pericytes are a clear source for MSCs isolated from various organs[21–25], and MSCs located in the cephalic region are of the neuroectodermal origin (via pericytes), but not the mesodermal origin[26].

Here, we hypothesize that MSCs can be generated from hPSCs also through the endodermal lineage. We show that, indeed endoderm progenitors can give rise to MSCs in vitro. Like MSCs from the other sources, the endoderm-derived MSCs also displayed anti-inflammatory activity and ameliorated dextran sulfate sodium (DSS)-induced colitis in ulcerative colitis mouse model.

## Results

**MSC generation from hPSCs via definitive endoderm progenitors.** To investigate whether MSCs could be induced from endoderm lineage in vitro, we decided to learn more about MSCs derived from endoderm organs. We examined the gene expression of MSCs derived from endoderm tissues. First, we derived MSCs from human endodermal tissues, including the colon and liver, and we also cultured MSCs originated from mesoderm adipose and extraembryonic umbilical cord as control (Supplementary Fig. 1a). We demonstrated that all these MSCs had the multipotency to become adipocyte, chondrocyte and osteocyte in cell culture (Supplementary Fig. 1b, c). We then compared the RNA-seq profile of these MSCs along with H9 hESCs. Based on transcriptome analysis, all the MSCs were clustered together against hESCs, and they all expressed genes that were specifically enriched in stromal cells according to Enrichr Cell Type Analysis (Supplementary Fig. 1d).

Next, we developed a method that could generate MSCs from hPSCs through definitive endoderm in cell culture. In order to track the emergence of definitive endoderm progenitors, SOX17 promoter-driven-GFP H9 hESCs were used in this project[27]. Five stages of treatments were applied on hESCs to induce MSCs through endoderm lineage (Fig. 1a). Stage 1 (day 0–1): Exit from pluripotency by WNT activation (GSK3 inhibitor CHIR99021) and Activin A in a defined medium; Stage 2 (day 1–3): The induction of definitive endoderm progenitors by Activin A in a defined medium; Stage 3 (day 3–5): MSC induction with MSC medium or serum-free condition; Stage 4 (day 5–15): MSC commitment with MSC medium or serum-free condition; Stage 5 (day 15–24): MSC enrichment and expansion in MSC medium.

During the MSC induction process, SOX17-GFP was not observed in hESCs on day 0, but most cells become SOX17-GFP positive at the end of Stage 2 on day 3, indicating the emergence of definitive endoderm progenitors. We also showed that most

cells were positive for both endoderm markers SOX17[27] and CXCR4[28], and the percentage of SOX17+/ CXCR4+ cells was above 90%. It suggested that definitive endoderm progenitors were effectively induced from hESCs in 3 days (Supplementary Fig. 2a, b). At stage 3, the culture medium was switched to MSC medium, and SOX17-GFP positive percentage was maintained on day 5, but they gradually decreased afterward (Fig. 1b). RT-qPCR assay showed that the expression of SOX17 started to decrease after day 3, while the expression of MSC classical markers, including CD44, NT5E (CD73) and ENG (CD105), were detected on day 15 (Fig. 1c). The expression of MSC markers further increased after subsequent passages (Fig. 1c). Flow cytometry analysis demonstrated that almost all cells became CD44+/ CD73+/CD105+, while remained CD45- after three passages (Fig. 1d). More detailed analysis showed that SOX17-GFP+ cells gradually decreased after day 3 (left panel of Fig. 1e) while cells with positive MSC markers increased (right panel of Fig. 1e). These data indicated that definitive endoderm progenitors could be induced to cells expressing classical MSC markers.

The proliferation ability of in vitro generated MSCs was confirmed (Fig. 1f). MSCs are usually defined by their multipotency to differentiate into multiple cell types in vitro. We treated the above endoderm-originated cells with specific differentiation conditions. We showed that those cells could be further induced into adipocytes, chondrocytes, and osteocytes (Fig. 1g, h). Considering that these cells fit the criteria of MSCs based on marker gene expression, morphology, and multipotency, we named them definitive endoderm-originated MSCs (DE-MSCs). Furthermore, we showed that DE-MSCs could be induced to pancreatic cell fate with the formation of an islet-like structure and the elevated expression of pancreatic progenitor marker genes PDX1[29] and PTF1A[30] (Supplementary Fig 2c and Fig. 2d). These results strengthened the argument that DE-MSCs had the multipotent potentials.

To confirm that DE-MSCs were indeed induced from definitive endoderm progenitors, we sorted SOX17-GFP positive DE progenitors on day 3 and cultured them in MSC medium (Supplementary Fig. 2e, f). SOX17-GFP DE progenitors survived well after sorting (Supplementary Fig. 2g), proliferated, and finally became fibroblast-like cells (Supplementary Fig. 2h). Flow cytometry demonstrated that those cells were positive for CD44, CD73, and CD105 (Supplementary Fig. 2i). These data confirmed that definitive endoderm progenitors could contribute to MSCs in cell culture.

To examine the efficiency of the DE-MSC induction protocol, we tried to generate DE-MSCs from the H1 hESC line and NL-1 hiPSC line. These cells were induced to definitive endoderm progenitors and were then treated in an MSC medium. Fibroblast-like cells were observed after three weeks with CD44+ and CD105+ expression (Supplementary Fig. 2j and Fig. 2k). We then demonstrated that these cells could differentiate into trilineage (Supplementary Fig. 2l). These results suggested that the DE-MSC induction method is robust for various hPSCs lines.

**Stepwise emergence of DE-MSCs in vitro revealed via scRNA-seq.** To understand how DE-MSCs emerge from definitive endoderm progenitors, scRNA-seq analysis was conducted after definitive endoderm was induced on day 3, day 5, and day 15. Based on the gene expression profile, cells were generally clustered according to their differentiation stages (Fig. 2a). On day 3 and day 5, most cells in these two stages were grouped together, respectively, indicating similar gene expression patterns. On day 15, two heterogeneous population emerge, implying cell type diversification (Fig. 2b). We showed that definitive endoderm marker genes (SOX17 and CXCR4) were highly expressed on day

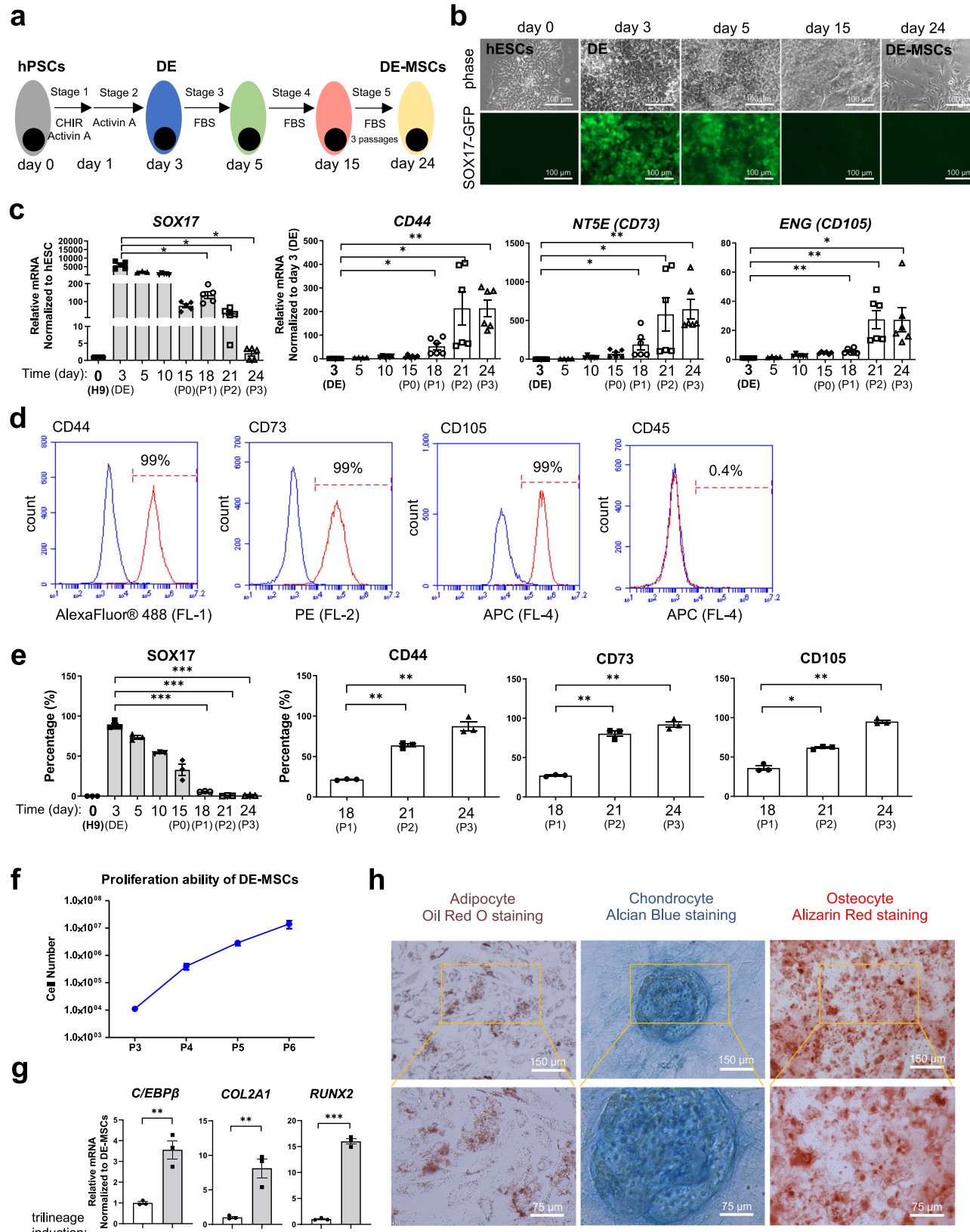

3, but they decreased afterward (Fig. 2c). Meanwhile, some other endoderm marker genes were also expressed on both day 3 and day 5, such as *FOXA2*[31], *OTX2*[31], *GATA4*[32], and *HHEX*[33] (Supplementary Fig. 3a). These results were consistent with Fig. 2 that most hESCs were effectively induced to definitive endoderm progenitors before MSC induction.

We then showed that some gut/liver axis genes were expressed on day 5, and were further elevated on day 15. Some gut marker genes, such as *TTR*[34], *AFP*[35], *APOA1*[36], *APOB*[36], and *H19*[37] were expressed in both clusters on day 15, and most of them had higher expression in cluster 1. The shared endoderm gene expression indicated that both clusters came from the definitive

**Fig. 1 MSC generation from hPSCs via definitive endoderm progenitors. a** Stage-wise differentiation strategy to induce DE-MSCs with definitive endoderm (DE) origin from hESCs. DE-MSCs were obtained on day 24. **b** Cell morphology (upper panel) and expression of SOX17-GFP (lower panel) from day 0 to day 24. Spindle-like cells were observed on day 24 and they were SOX17-GFP negative on day 24. Scale bar = 100 μm. **c** RT-qPCR analysis for mRNA level of *SOX17*, *CD44*, *NT5E (CD73)*, and *ENG (CD105)* on day 3, 5, 10, 15 (Passage 0, P0), 18 (Passage 1, P1), 21 (Passage 2, P2), 24 (Passage 3, P3) (*n* > 3), *$p < 0.05$, **$p < 0.01$. **d** Flow cytometry analysis of percentage of CD44$^+$, CD73$^+$, CD105$^+$, CD45$^+$ cells. The blue line represented IgG isotype control for the gating strategy, the red line represented the percentage of CD44$^+$, CD73$^+$, CD105$^+$, and CD45$^+$ cells. **e** Flow cytometry analysis of the percentage of SOX17$^+$ cells on day 0, 3, 5, 10, 15, 18 (P1), 21 (P2), 24 (P3) (left panel), and flow cytometry analysis of the percentage of CD44$^+$, CD73$^+$, and CD105$^+$ cells on day 18 (P1), 21 (P2), 24 (P3) (right panel) (*n* = 3). *$p < 0.05$, **$p < 0.01$, and ***$p < 0.001$. **f** Total cell number of DE-MSCs from passage 3 (P3) to passage 6 (P6) (*n* = 3). **g** RT-qPCR analysis was used to detect adipogenic (*C/EBPβ*), chondrogenic (*COL2A1*) and osteogenic (*RUNX2*) marker genes (*n* = 3), **$p < 0.01$, ***$p < 0.001$. **h** DE-MSCs were induced into adipocytes (stained with Oil Red O), chondrocytes (stained with Alcian Blue), and osteocytes (stained with Alizarin Red). Scale bar = 150 μm for an upper panel, scale bar = 75 μm for lower panel.

endoderm progenitors. Meanwhile, stromal genes[38] (*COL1A1*, *LOX*, *COL3A1*, *CCDC80*, and *THY1*) were highly expressed on day 15, especially in cluster 2 (Fig. 2d and Supplementary Fig. 3a). The current evidence suggested that MSCs emerged with some endoderm gene expression.

We then analyzed the gene expression profiles according to differentiation stages. Cells from day 3 and day 5 had more uniform gene expression profiles, while cells from day 15 could be split into two clusters (cluster 1 and 2). Enrichr cell type analysis shows that cluster 1 were more associated with hepatocyte, liver (bulk tissue), and gastric tissue (bulk), while cluster 2 were associated with stromal cell type fibroblast (Fig. 2e). We then demonstrate that cells in cluster 2 had higher expression of MSC markers such as *CD44*, *NT5E (CD73)*, *ENG (CD105)*, and *NGFR (CD271)* (Fig. 2f). These scRNA-seq results indicated that DE-MSCs emerged from definitive endoderm between day 5 and day 15.

We further examined DE-MSC induction under mitogenic regulation from day 3 to day 5. WNT pathway activator CHIR99021 (CHIR)[39] significantly elevated the expression level of *CD44*, *NT5E (CD73)*, and *ENG (CD105)*. TGFβ / Activin pathway inhibitor SB431542[40] and CHIR together also promoted MSC markers. In contrast, the combination of WNT inhibitor XAV939[39] and SB431542 significantly suppressed MSC gene expression (Supplementary Fig. 4a). Further study showed that DE-MSCs can be induced under CHIR or CHIR/SB short-term treatment (Supplementary Fig. 4b). And these DE-MSCs could also be induced to adipocytes, chondrocytes and osteocytes (Supplementary Fig. 4c, d). These results indicate that WNT and TGFβ pathways play important roles in DE-MSC induction.

Till now, all DE-MSCs were induced with an MSC medium that contained bovine fetal serum (FBS), so we examined whether MSCs could be induced under serum-free conditions (Supplementary Fig. 5a). In serum-free chemically defined medium, the expression of MSC marker genes was elevated on day 15 (Supplementary Fig. 5b). MSC differentiation was enhanced by WNT activation (CHIR99021) and CHIR99021/SB431542 treatment (Supplementary Fig. 5c). scRNA-seq also showed that stromal population emerged without serum (Supplementary Fig. 5d). However, those cells from serum-free conditions could not be effectively expanded (data is not shown), suggesting that more studies are necessary to expand DE-MSCs in serum-free conditions.

**Expression of origin-specific markers in DE-MSCs.** In order to evaluate whether DE-MSC induction history could lead to any molecular signatures in MSCs, we compared DE-MSCs with MSCs that were induced from other lineages, including mesoderm-originated MSCs (Meso-MSCs)[41], neural crest-originated MSCs (NC-MSCs)[42], and trophoblast-originated MSCs (Troph-MSCs)[43] (Fig. 3a). The lineage specificity was demonstrated by the expression of specific progenitor marker genes such as *MESP1* (mesoderm), *SOX10* (neural crest), and

*CGB* (trophoblast) (Supplementary Fig. 6a). Meso-MSCs, NC-MSCs, and Troph-MSCs were then induced in MSC medium, and they were CD44$^+$ and CD105$^+$ (Supplementary Fig. 6b). We further demonstrated that all these cells had trilineage differentiation capacity (Supplementary Fig. 6c, d).

We then compared the transcriptomes of hESCs, DE-MSCs, Meso-MSCs, NC-MSCs, and Troph-MSCs. Compared to hESCs, all MSCs expressed genes that were enriched in fibroblast cell types by Enrichr analysis (Fig. 3b). DE-MSCs were clustered together, and NC-MSCs and Troph-MSCs were more closely associated. Further comparison among MSCs showed that each MSC type expressed genes associated with the specific lineage that were associated with the induction methods. DE-MSCs expressed genes enriched in endoderm organs, including the intestine, lung, and stomach. Meso-MSCs were related to myofibroblast and cardiovascular cells, NC-MSCs were associated with the fetal brain and spinal cord, and Troph-MSCs were related to the human placenta (Fig. 3c). We then showed that DE-MSCs derived with FBS alone were more associated with omentum and lung, CHIR treatment drove gene expression to more related to intestine, while CHIR/SB dual treatment led to upregulation of colon and pancreas related genes (Fig. 3d). These results indicated that MSC differentiated from endoderm progenitors could lead to endoderm signatures.

**DE-MSC-mediated modulation of inflammatory responses in cell culture and mouse model.** We next examined the immunoregulatory ability of DE-MSCs. MSCs were first exposed to proinflammatory factors IFN-γ, and we then checked the expression of proinflammatory cytokines (*IL-6*, *IL-8*, and *CCL2*) and anti-inflammatory cytokines (*IDO1*, *PD-L1*, and *TSG6*). RT-qPCR showed that IFN-γ upregulated the expression of proinflammatory *IL-6*, *IL-8*, and *CCL2* in both DE-MSCs and UC-MSCs. For the anti-inflammatory genes, IFN-γ increased the expression of *IDO1* and *PD-L1* in both DE-MSCs and UC-MSCs, but *TSG6* expression was not significantly changed (Fig. 4a and Supplementary Fig. 7a). We also showed that DE-MSCs (FBS) could upregulate anti-inflammatory cytokine TGF-β production in spleen cells (Fig. 4b). These findings suggested that DE-MSCs might have the potential to modulate inflammatory response through multiple pathways.

We then examined the effect of DE-MSCs in DSS-induced colitis in mice, which resembles human ulcerative colitis[44]. DSS treatment caused a significant decrease in weight in 14 days (Fig. 4c). However, peritoneal injection of DE-MSCs significantly alleviated the loss of body weight. After DE-MSC treatments, the body weight recovered close to that of a healthy animal after 14 days. We then demonstrated that DSS shortened colon length, but the symptom was suppressed by DE-MSCs (Fig. 4d, e). The analysis of the distal colon sections demonstrated that DSS disrupted intestinal epithelium, but DE-MSCs helped to suppress the disruption (Fig. 4e). H&E staining showed that the mouse

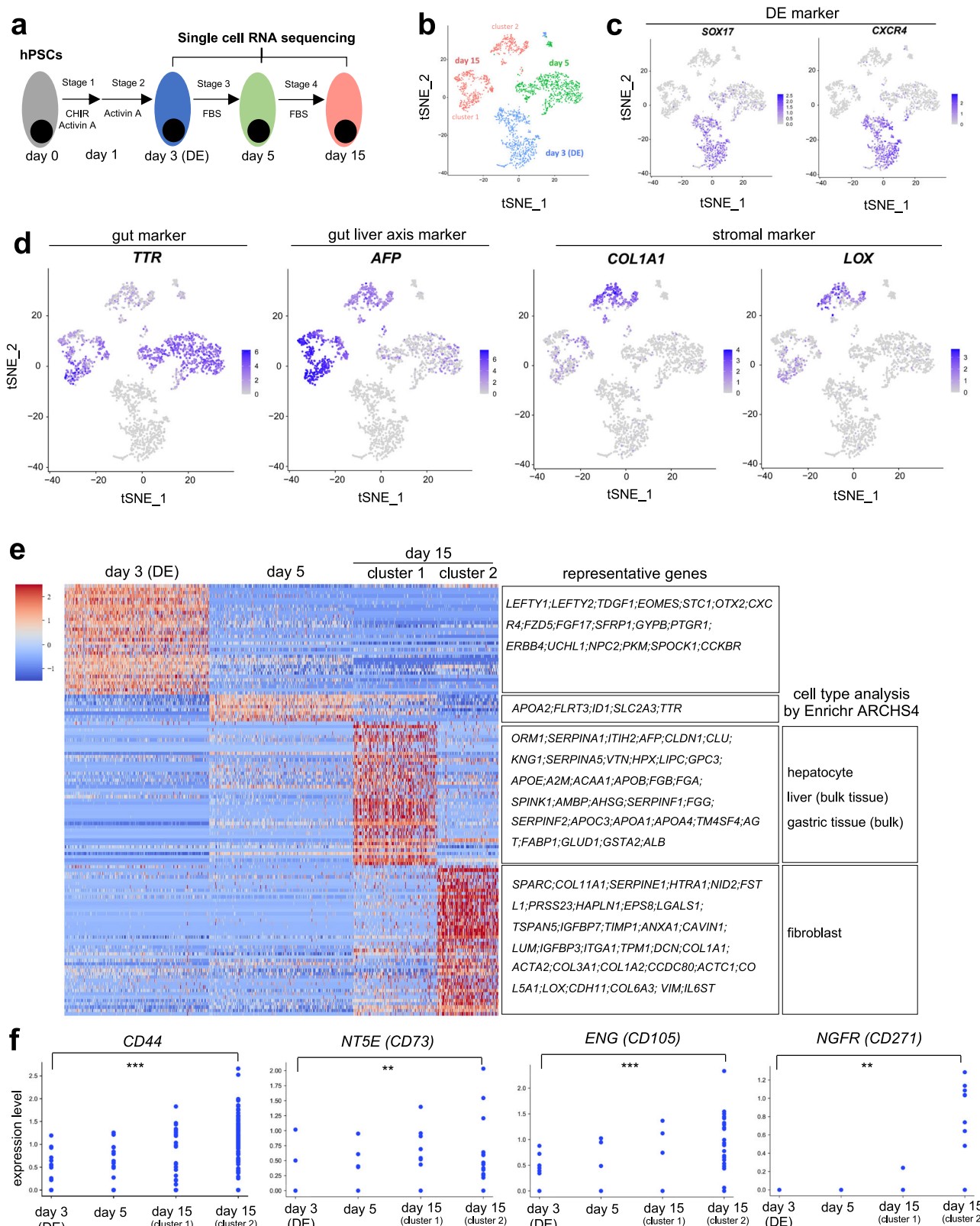

COMMUNICATIONS BIOLOGY | (2023)6:476 | https://doi.org/10.1038/s42003-023-04810-5 | www.nature.com/commsbio

intestinal epithelium was disrupted under DSS treatment, but DE-MSCs (FBS), DE-MSCs (CHIR), or DE-MSCs (CHIR/SB) treatments ameliorated epithelium disruption (Fig. 4f). Another pathological manifestation of colitis was an increase of CD8[+] cytotoxic T cells in colon mucosa[45,46]. The infiltration by CD8[+] cytotoxic T cells in DSS-induced colitis mice was detected by immunostaining. DE-MSCs and UC-MSCs treatments significantly reduced CD8[+] cytotoxic T-cell infiltration (Supplementary Fig. 7b). One pathological manifestation of colitis is the depletion of mucin-producing goblet cells[47]. AB/PAS staining demonstrated the loss of mucin-producing goblet cells (rose red color cells) in DSS-induced colitis mice, but DE-MSCs and UC-MSCs

**Fig. 2 Stepwise emergence of DE-MSCs in vitro revealed via scRNA-seq. a** Schematic figure showing the time points for collecting samples for single-cell RNA sequencing. **b** tSNE projection of day 3 (DE), day 5, and day 15 samples, and there were two clusters (cluster 1 and 2) in day 15 sample. **c** tSNE projection of definitive endoderm marker gene (*SOX17* and *CXCR4*) expression on day 3 (DE), day 5, and day 15 clusters, indicating that *SOX17* and *CXCR4* were predominantly expressed in day 3 (DE) samples. **d** tSNE projection of gut (*TTR*), gut liver axis (*AFP*), and stromal (*COL1A1* and *LOX*) marker gene expression on day 3 (DE), day 5, and day 15 clusters, showing that cells in cluster 2 of day 15 sample predominantly expressed *AFP*, *COL1A1*, and *LOX*. **e** Heatmap showing the column-scaled expression of DEG per cluster in day 3 (DE), day 5, and day 15 samples. Representative genes in each cluster were also shown. In addition, cell types were identified based on Enrichr. **f** Scatter plots showing the gene expression levels of *CD44*, *NT5E* (*CD73*), *ENG* (*CD105*), and *NGFR*(*CD271*) on day 3 (DE), day 5, and day 15 (cluster 1 and cluster 2) at single-cell resolution, indicating the emergence of MSCs in cluster 2 of day 15 sample.

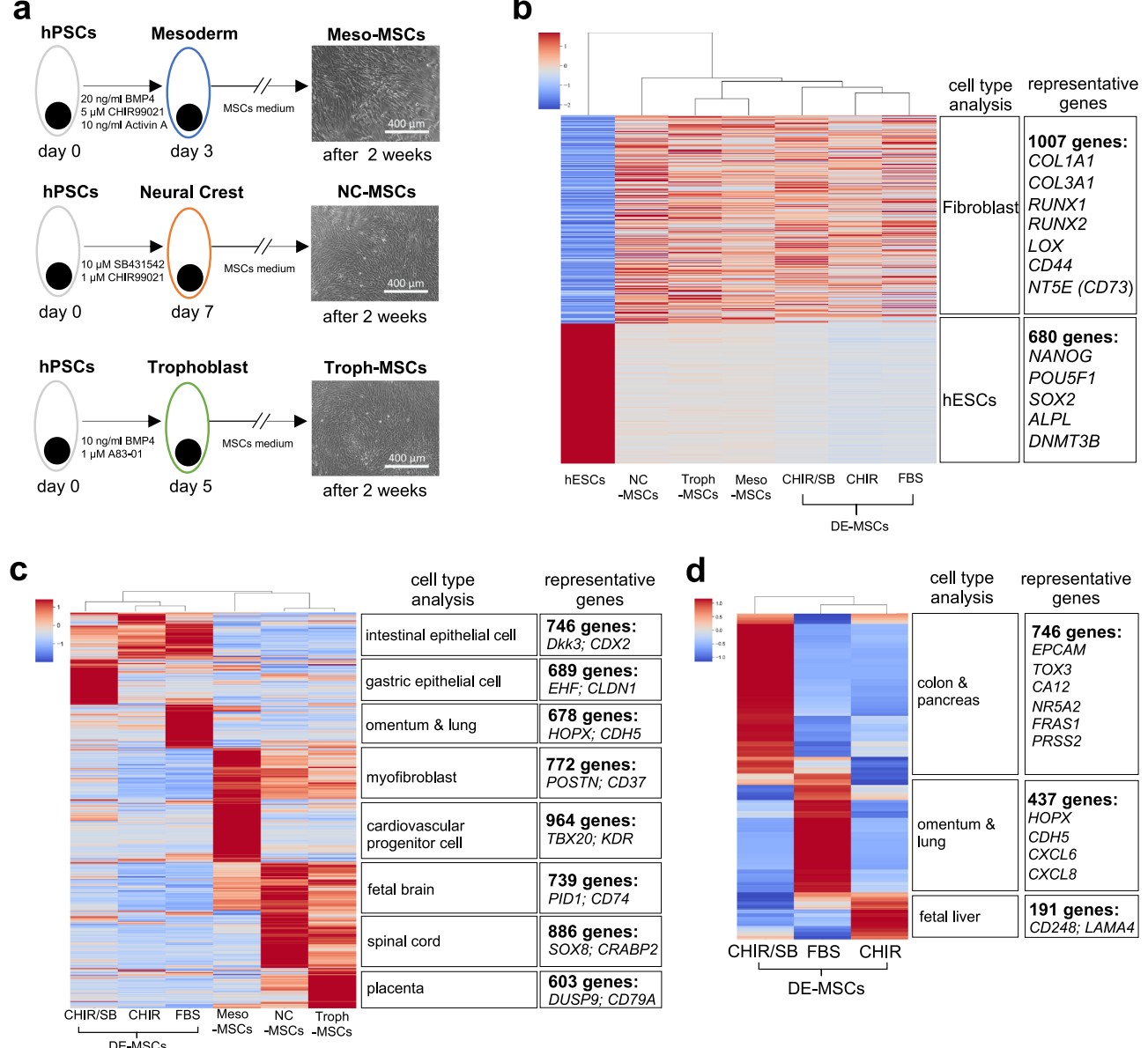

**Fig. 3 Expression of origin-specific markers in DE-MSCs. a** Schematic flow of in vitro MSC generation from mesoderm (Meso-MSCs), neural crest (NC-MSCs), and trophoblast (Troph-MSCs) origins, scale bar = 400 μm. **b** Heatmap of DEG in NC-MSCs, Troph-MSCs, Meso-MSCs, DE-MSCs (CHIR/SB), DE-MSCs (CHIR), and DE-MSCs (FBS) compared with hESCs and cell type prediction by Enrichr database, indicating that all these MSC highly expressed the genes related to fibroblast rather than hESCs. **c** Heatmap of DEG in DE-MSCs (CHIR/SB), DE-MSCs (CHIR), DE-MSCs (FBS), Meso-MSCs, NC-MSCs, and Troph-MSCs and cell type prediction by Enrichr database, showing that MSCs originated from different developmental origin has organ-specific gene expression signatures. **d** Heatmap of DEG in DE-MSCs (CHIR/SB), DE-MSCs (FBS), and DE-MSCs (CHIR) and cell type prediction by Enrichr database, indicating CHIR or CHIR/SB treatment from day 3 to day 5 modified the gene signatures of DE-MSCs.

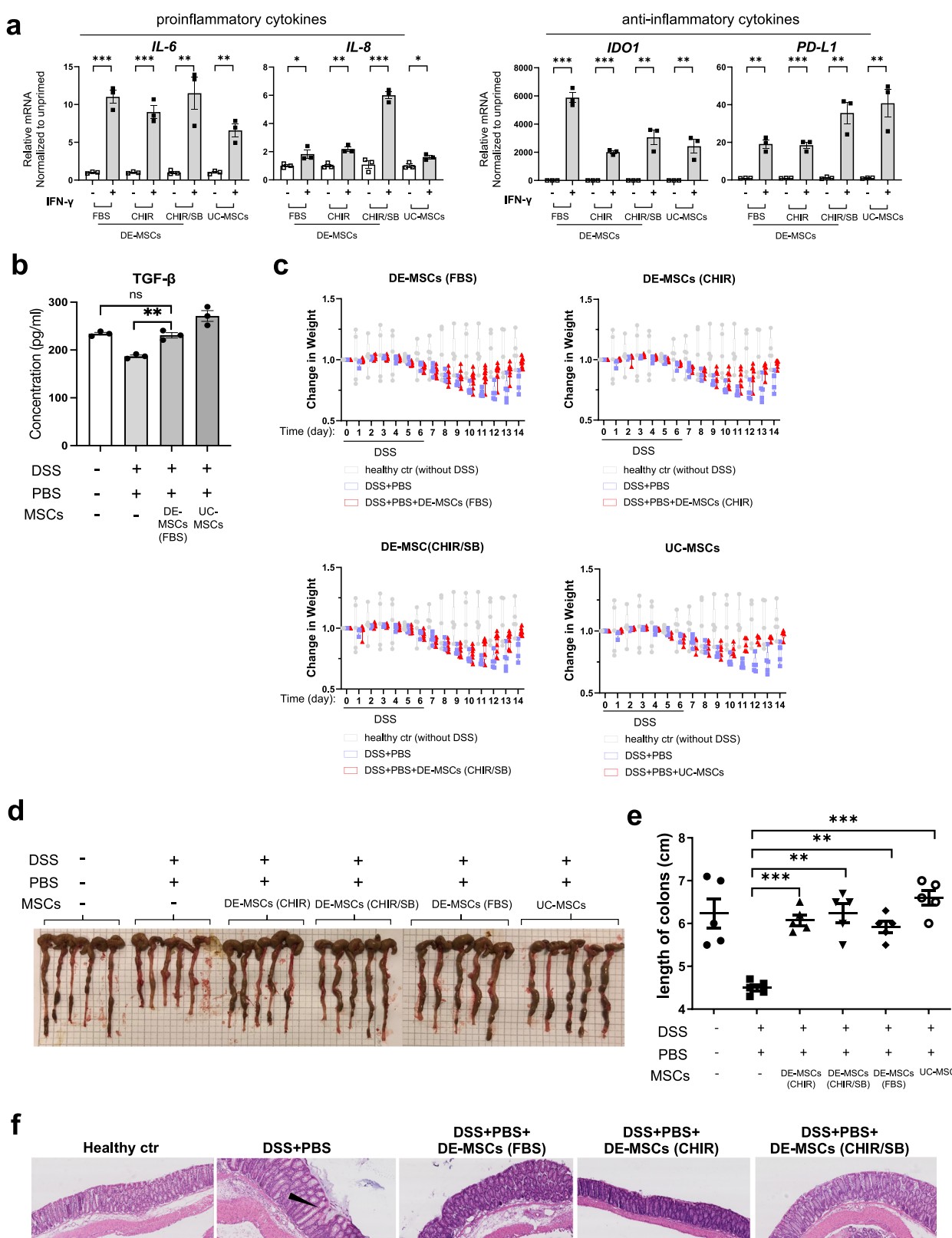

treatments rescued the phenotype and ameliorated pathological deterioration (Supplementary Fig. 7c). According to previous studies[48,49], DE-MSCs might have brought beneficial effects through their interaction with intermediate cells in DSS-induced colitis model. These data suggested that DE-MSCs could have therapeutic effects on colitis.

## Discussion

MSCs are, in essence, cultured cells isolated from various tissues such as the bone marrow, skeletal muscles, placenta, and pancreas and have demonstrated remarkable effects in immune regulation and tissue regeneration as a therapy[21,22]. Whether identical cells exist naturally in vivo is unclear. Presumptive MSCs exist in

**Fig. 4 DE-MSC-mediated modulation of inflammatory responses in cell culture and mouse model. a** mRNA levels of proinflammatory cytokines (*IL-6* and *IL-8*) and anti-inflammatory cytokines (*IDO1* and *PD-L1*) were analyzed by RT-qPCR assay in DE-MSCs (FBS), DE-MSCs (CHIR), DE-MSCs (CHIR/SB), and UC-MSCs that were exposed to IFN-γ (*n* = 3). \*p < 0.05, \*\*p < 0.01, \*\*\*p < 0.001. **b** Effect of DE-MSCs (FBS) and UC-MSCs on anti-inflammatory cytokine TGF-β level in spleens of colitis mice, showing the anti-inflammatory effect of DE-MSCs (FBS) (*n* = 3). \*\*P < 0.01. **c** Average changes in body weight of each group from day 0 to day 14 (3–5 mice in each group). The unit of body weight was the gram. **d** Images of colons dissected from mice in healthy mice, DSS + PBS, DSS + PBS + DE-MSCs (FBS), DSS + PBS + DE-MSCs (CHIR), DSS + PBS + DE-MSCs (CHIR/SB), and DSS + PBS + UC-MSCs groups on day 14 (*n* = 5). **e** The length of colons from multiple experiments described in (**c**) were measured and plotted (*n* = 5). \*\*p < 0.01, \*\*\*p < 0.001. **f** Hematoxylin and eosin (H&E) staining of distal colon sections from healthy mice, DSS + PBS, DSS + PBS + DE-MSCs (FBS), DSS + PBS + DE-MSCs (CHIR), DSS + PBS + DE-MSCs (CHIR/SB) groups, scale bar = 100 μm.

various tissues and organs, however, the exact role of which in tissue turnover and repair is not known. MSCs have been induced from hPSCs through the mesoderm, neural crest, and extra-embryonic lineages[15–18]. However, it is unclear whether the definitive endoderm has the potential to contribute to MSCs. In this study, we have demonstrated that definitive endoderm progenitors can be induced to generate MSCs (DE-MSCs) which possess multipotency and therapeutic effects.

Definitive endoderm progenitors have the potential to generate endodermal cell types such as hepatocytes, pancreatic islet cells, and pulmonary cells, given specific signaling modulators at proper times. The cell fate specification procedure can be diverted by exposing the progenitors to an MSC medium containing FBS, which leads to the emergence of MSCs. Apparently, some serum factors promote MSC fate against other more specialized endoderm cell types. We show that the induction of DE-MSCs is promoted by WNT activation but suppressed by WNT inhibition. Interestingly, WNT activation is also found to promote mesenchymal cell types related to epicardial and neural crest cells, which is consistent with previous reports[50,51]. It is possible that cells of all lineages have the plasticity to become MSCs before cell fate is finally determined. WNT activation is partially responsible for switching cell fate towards MSCs during organ formation[52–54].

It has been reported that perivascular cells contribute to the isolation of MSCs from endodermal organs[21,55,56]. Perivascular cells, including pericytes and adventitial cells, have proved as native sources of MSCs in developing and adult human organs[57]. However, it awaits to be proved whether endodermal cells can be converted to MSCs in situ. In summary, this study demonstrates that definitive endodermal progenitors are a source of MSCs, which may be used for the treatment of degenerative diseases, especially, those in endodermal organs.

## Methods

**hPSC maintenance and differentiation to DE-MSCs.** Under approval of the University of Macau's Research Ethics Board, hESC lines H1 and H9 (from WiCell Research Institute, Inc., Madison, WI, http://www.wicell.org) and hiPSC line NL-1 (from NIH) were cultured in Matrigel-coated six-well plates in E8 medium with daily change, and cells were passaged every 3–4 days with EDTA when they reached 60–70% confluence[58–60]. For differentiation, 1 × 10⁴ cells per well were seeded onto Matrigel-coated 24-well plate, and differentiation was initiated when cells reached about 40–50% confluence. hPSCs were induced into mesoendoderm under 5 μM CHIR99021 and 100 ng/ml Activin A in differentiation medium (DMEM/F12 supplemented with transferrin, chemically defined lipid concentrates, ascorbic acid, and sodium selenite) for 1 day, and definitive endoderm progenitors were patterned under 100 ng/ml Activin A treatment in differentiation medium for another 2 days. Subsequently, definitive endoderm progenitors were further differentiated in αMEM supplemented with 1 × MEM NEAA, 1 × GlutaMAX™-I, 100 ng/ml FGF2, 10 μg/ml insulin, and 20% FBS (FBS condition) or without 20% FBS (serum-free condition) for another 12 days, and DE-MSCs (FBS condition) were generated after 2–3 passages (6–9 days). Optimally, 5 μM CHIR99021, 10 μM XAV939, 10 μM SB431542, and their combination can be applied from day 3 to day 5 under FBS condition or serum-free condition, and DE-MSCs (CHIR and CHIR/SB condition) can also be generated after 2–3 passages (6–9 days). FBS condition medium or serum-free condition medium was changed every 2 days. The information of chemicals and recombinant proteins was listed in Supplementary Table 1.

**Isolation and expansion of MSCs from various human tissues.** All primary cells were derived with ethics approval from Zhuhai People's Hospital and the University of Macau. Colon-MSCs, liver-MSCs, and adipose-MSCs were isolated from the non-tumor colon, liver, and breast tissues, respectively. Umbilical cord-MSCs are isolated from umbilical cord tissue. The isolation and expansion of primary mesenchymal stem cells followed these steps: Briefly, the human tissues were cut into small pieces, and were then treated in a series of solutions at 37 °C, including Collagenase solution (DMEM/F12 supplemented with 5% FBS, 5 μg/ml insulin, 500 ng/ml hydrocortisone, 10 ng/ml EGF, 20 ng/ml cholera toxin, 300 U/ml collagenase III, 100 U/ml hyaluronidase) with intermittent pipetting for 2 h, Dispase solution (DMEM/F12 supplemented with 5 mg/ml Dispase II and 0.1 mg/ml deoxyribonuclease) for 5 min, and 0.25% trypsin solution for 2 min. The digestion was then stopped by DMEM/F12 medium with 5% FBS, and cells were washed with Hanks solution before treatment with RBC lysis buffer. After cells were washed with Hanks solution, they were cultured in a cell culture plate with MSC media. Cells were passaged 5 times before they were harvested for gene expression and functional analysis[61].

**Pancreatic induction from DE-MSCs.** DE-MSCs were induced to pancreatic fate following these steps: Briefly, DM-MSCs were first cultured till 80% confluence, and were then differentiated towards β-cell-like cells in DMEM-low glucose medium, containing 5% platelet lysate, 10 μM retinoic acid (for 24 h only), 100 ng/ml Activin, 200 ng/ml glucagon-like peptide I (GLPI-1), 20 ng/ml epidermal growth factor (EGF), 10 ng/ml fibroblast growth factor (FGF), 10 ng/ml β-cellulin, 10 mM nicotinamide, and 2 mM glutamine. Cells were cultured in adherent conditions for the first 7 days and were then changed to ultra-low attachment dishes for another 2 weeks[62].

**Trilineage differentiation of DE-MSCs.** For osteogenic differentiation, DE-MSCs were cultured in high glucose DMEM containing 10% FBS, 1×L-glutamine, 0.1 μM dexamethasone, 10 mM β-glycerophosphate, 10 ng/ml BMP2, and 50 μg/ml ascorbic acid for 21 days with medium changes every 3 days. For chondrogenic differentiation, DE-MSCs were cultured in high glucose DMEM, 10% FBS, 2 mM L-glutamine, 0.1 μM dexamethasone, 50 μg/ml ascorbic acid, 1% sodium pyruvate, 10% insulin-transferrin-selenium, 10 ng/ml BMP2, and 10 ng/mL TGFβ for 21 days with medium changes every 3 days. For adipogenic differentiation, DE-MSCs were cultured in high glucose DMEM containing 10% FBS, 2 mM L-glutamine, 1 μM dexamethasone, 0.5 mM isobutylmethylxanthine, 50 μM indomethacin, and 10 μg/ml insulin for 21 days with medium changes every 3 days[63–65].

**Generation of MSCs from hPSCs via the mesoderm, neural crest, and trophoblast origins.** The generation of mesoderm, neural crest, and trophoblast progenitors were first induced according to published protocols with minor modifications[41–43]. These progenitors were then cultured in an MSC medium for 14 days and were subsequently passaged three times to generate mesoderm-originated MSCs (Meso-MSCs), neural crest-originated MSCs (NC-MSCs), and trophoblast-originated MSCs (Troph-MSCs).

**A mouse colitis model induced with DSS.** Mice were administered with 2% DSS (molecular weight 36,000–50,000, MP Biomedical) in the drinking water for 6 days to establish the colitis mouse model[18]. Each mouse was injected intraperitoneally with 5 × 10⁶ DE-MSCs (FBS, CHIR, or CHIR/SB condition) or 5 × 10⁶ UC-MSCs in 1 × PBS or 1 × PBS alone (negative control) on day 2 and day 3 after the start of the DSS treatment. The body weight of mice were measured daily from day 0 to 14. On day 14, these mice were euthanized by CO₂ asphyxiation.

Upon necropsy, each mouse's colon was dissected and measured for its length. The colons were rinsed with sterile 1 × DPBS and fixed in 4% PFA at 48 °C for 48–72 h. The distal part of the colon was embedded in paraffin wax and sectioned at 5 μm in thickness, mounted to glass slides, and hematoxylin and eosin (H&E) staining and immunostaining with PE-conjugated Rat Anti-Mouse CD8a antibody (BD Biosciences, cat. No. 553032) were performed. In addition, Alcian blue/Periodic Acid-Schiff (AB/PAS) staining was used to detect mucin-producing goblet cells according to the manufacturer's protocol (Sigma-

Aldrich, 395B-1KT). Images of stained sections were acquired on Leica whole slides scanner SCN400F.

**RT-qPCR.** mRNA was extracted with RNAiso-plus (TAKARA, cat. No.108-95-2) and reverse transcription (RT) was performed with a High-Capacity cDNA Reverse Transcription kit (Applied Biosystems, cat. No. 4368813). qPCR was conducted with SYBR Premix Ex Taq (TAKARA, cat. No. RR420) and the Quantstudio-7 system (Applied Biosystems). The relative amounts of the amplified nucleotide fragment were calculated by the $2^{\wedge}(-\Delta Ct)$ method. Primer sequences were listed in Supplementary Table 2.

**Flow cytometry.** The MSCs were evaluated according to the expression of CD44, CD73, CD105, and PDGFRβ by flow cytometer (Becton Dickinson C6). The antibodies used were CD44 (156-3C11) Mouse mAb (1:1000, CST), anti-CD73 antibody (ab54217) (1:1000, Abcam), anti-CD105 antibody (ab11414) (1:1000, Abcam), and anti-PDGFR β antibody (ab69506) (1:1000, Abcam).

**ELISA assay for anti-inflammatory cytokines.** Monocytes were isolated from spleens and cultured in RPMI 1640 medium ($1 \times 10^6$ cells/well in a 24-well plate). The supernatants were collected 3 days later for ELISA assay[66]. The level of anti-inflammatory cytokine TGF-β was measured by a murine ELISA kit according to the manufacturer's protocol (Solarbio).

**scRNA-seq.** Single cells were isolated and processed with the Nadia system (Dolomite Bio), and the sequencing libraries were prepared according to standard protocol[67].

**Bulk RNA-seq.** Total mRNA was extracted by RNAiso-plus (TAKARA, cat. No.108-95-2). The RNA libraries were generated using the TruSeq RNA Sample Preparation kit (Illumina), and cDNA fragments were enriched by PCR using Illumina TruSeq PCR primers. Each library was sequenced as paired-end reads in HiSeq 2000/1000 (Illumina).

**Bioinformatic analysis.** The raw data of scRNA-seq was processed by the Drop-seq tools protocol (http://mccarrolllab.org/). Data normalization was conducted by R package "Seurat". Seven hundred cells were chosen for each group and 2000 variable features were picked to cluster. The tSNE graphs for the FBS condition were set up according to the first 14 PCs (k = 20). The tSNE graphs for the serum-free condition were set up according to the first 13 PCs (k = 20). Cell types were identified with R package "SingleR" and Enrichr (https://maayanlab.cloud/Enrichr/).

Transcripts Per Million (TPM) was used to normalize gene read counts for bulk RNA-seq analysis. TPM values of corresponding genes were further normalized based on Z-score. R package ggplot2, as well as the Python package seaborn, was used to generate heatmaps. R package Deseq2 was used to pick differentially expressed genes (DEG) with q value <0.05, fold change >2 or <−2.

**Statistics and reproducibility.** The standard error of the mean (SEM) was calculated based on three or more independent experiments. Statistical significance was determined using a t-test.

**Data transparency.** The web-based tool Enrichr is available at https://maayanlab.cloud/Enrichr/.

**Reporting summary.** Further information on research design is available in the Nature Portfolio Reporting Summary linked to this article.

## Data availability
The scRNA-seq and bulk RNA-seq data of this study have been deposited in the NCBI's BioProject under accession code PRJNA709641. The numerical source data for RT-qPCR, flow cytometry, mouse weight, and mouse colon length are available in Figshare (https://doi.org/10.6084/m9.figshare.22229827). All other data are available from the corresponding author on reasonable request.

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

## Acknowledgements

We want to thank the technical support from the Bioimaging and Stem Cell Core Facility and Animal Facility at the University of Macau. We also thank the council members of the Macau Society for Stem Cell Research (MSSCR) for the constructive discussions. This project was funded by the University of Macau (File No. MYRG2018-00135-FHS and MYRG2019-00147-FHS) and by the Science and Technology Development Fund, Macau SAR (File No. 131/2014/A3, 056/2015/A2, 0059/2019/A1, 0123/2019/A3, 0011/2019/AKP, and 0002/2021/AKP).

## Author contributions

Y.Z., R.-H.X., and G.C. designed and planned the project; Y.Z. and W.L. conducted stem cell maintenance, differentiation, and fluorescence-activated cell sorting; Y.Z. characterized the DE-MSCs; Y.Z. and L.H. conducted the scRNA-seq on the Nadia instrument; Y.Z., X. Xiao, J.X., N.S., U.I.C., and E.C.W.C. conducted bioinformatics analysis; Y.Z., D.Z., and Y.Y. evaluated the therapeutic function of DE-MSCs on colitis mice model; Y.Z., Y.M., L.L., X. Xu, R.-H.X., and G.C. designed the experiments and analyzed data; Y.Z., H.C.K., W.L., and G.C. wrote and revised the manuscript. All authors read and approved the final manuscript.

## Competing interests

The authors declare no competing interests.

## Ethics approval

The use of hPSCs was approved by the University of Macau. The use of donor tissues and cells was approved by Zhuhai People's Hospital's Research Ethics Board.
