## [Peer Review File · Communications Biology]

Reviewers' comments:

Reviewer #1 (Remarks to the Author):

The report by Zhang and collaborators is two-fold. One part describes the development in vitro of MSC (mesenchymal stem cell) like cells from human pluripotent cells experimentally committed in culture to endoderm development. This is an interesting, original observation, although more experiments are required before such an unexpected transition in culture can be validated (see detailed comments below). In parallel, the authors propose that these observations reflect the direct contribution, in vivo, of the endodermal germ layer to the emergence of a subset of specialized MSCs retaining expression of endodermal tissue markers. Data provided in support of this hypothesis are indirect and insufficient. Much more would be needed to make a convincing case.

The manuscript is sometimes challenging to follow, due to the abundance of data presented (multiple conditions used in induction cultures, for instance, with no significant differences observed: see Fig4). The report would benefit from more concision and focus.

Besides experimental results, a striking observation is that authors have ignored, in the introduction and discussion, directly relevant data produced by other groups. The fact that MSCs can be grown in culture from virtually all organs and tissues has been interpreted as reflecting their native association with blood vessels. Perivascular cells sorted to homogeneity – including from organs of partial endodermal origin like the pancreas and kidney – yield MSCs in culture, although the contribution of differently distributed progenitors cannot be formally excluded. This attributes a mesodermal origin to most MSCs, with the exception of pericytes from the cephalic region which originate in the neural crest and retain MSC potential in vitro. All these results, which have been the object of multiple publications, should have been discussed in the context of the present report.

Main other comments

- Introduction: "Mesenchymal stem/stromal cells (MSCs) are multipotent stem cells that widely exist in diverse tissues and organs in human body". Discussion: "MSCs are integral components of human body, and their heterogenous presence provides essential supports to various functions in different organs". Where is the evidence for these statements? While this is an interesting possibility, this reviewer is not aware of any published experiment that supports this contention. MSCs are, by definition, long-term cultured cells. Evidence that a functionally identical cell (multipotent, immunosuppressive...) naturally exists in vivo is still missing. All tissues contain cells which, upon culture, give rise to MSCs; that these innate cells be identical to their in vitro descent is not yet proven, albeit possible. The authors should be prudent, not readily extrapolate results obtained in culture, and moderate this statement
- For how many passages were MSCs from pre- and postnatal tissues maintained prior to experimental use (Fig.1)? This is an important information with respect to cellular heterogeneity, which could not be found
- Regarding the point above, gene expression analysis on total MSC populations is difficult to interpret, with respect to the sensitivity of the techniques used and since these cultures are derived from total, unselected cell suspensions. Obviously, unrelated cells persist in these heterogeneous cell cultures for a while, notwithstanding the possibility of cell fusion in these conditions. The authors write "These data suggested that MSCs from different organs could retain tissue-specific expression". The fact that MSCs from different organs may have some distinct properties is actually supported by many other works, but it will take more than the data presented here to demonstrate that these differences reflect distinct germ line origins
- The relevance to MSCs of data obtained by scRNAseq on "stromal" cells present in prenatal tissues and the adult rectum is very unclear... Please explain
- The kidney and thymus are described as mesodermal; these are in fact endo-mesodermal organs. In addition, the mesoderm that contributes to thymus development is of ectodermal (neural crest) origin (so called mesectoderm), as precisely documented in animal chimaeras
- It should be helpful to get a closer look at the morphology of cells in transition to MSCs (higher

magnification, Fig.2b)

- Did the authors check whether their DE derived MSCs retain potential to differentiate into endodermal tissues?
- Fig.2g: some stainings are sparse and higher magnifications of framed areas are not very informative. How long did cells cultured in differentiation media? Was the expression of genes activated in these three lineages monitored in parallel?
- The observed emergence of MSC like cells from Sox17+ cells is interesting and original. However, not all DE cells exhibit bright Sox17 nuclear expression at d3 (supplem. Fig2). Also, the gate used to sort Sox17+ cells by FACS is not shown; did it include GFPlo cells? Did the authors verify that some early induced mesendoderm cells did not resist activin A induction, failing to differentiate into endoderm? Did they check the presence of cells expressing mesoderm markers (brachyury...) at successive times in culture? These are important controls, as the persistence of mesoderm cells might explain the emergence of MSCs in these cultures. The authors state indeed that "most hESCs became definitive endoderm progenitors". The issue of the role of distinct signalings in early MSC induction has been examined in the paragraph "Signal modulations affected DE-MSc induction", but in a rather confusing way. The authors are encouraged to describe these data more comprehensively
- Fig. 5 describes MSCs derived from either "mesoderm", a neural crest intermediate, or trophoblast, and illustrates expression by these cells of markers of diverse cell lineages representative of the respective germ lines of origin. However, there is no description at all of the methods and tissues used to derive these MSCs, culture conditions, number of passages, phenotype... as mentioned earlier, transcriptome results may be tainted by contaminating cells from tissues of origin. No interpretation is possible with so little information. This part of the work should be thoroughly re-written
- English spelling and style are of uneven quality; the main text is overall well written, while figure legends include obvious statements and unnecessary repeats but often lack important information

Minor points

- Pages/lines should be numbered, which should make it much easier for reviewers to write comments
- Paragraph titles in the Results section and Figure Legend should be written in present tense, not past: it gives the impression of one-time results that may not be reproduced...
- CD44, CD73, and CD105 are not "MSC specific markers" (Results, paragraph 3)

Reviewer #2 (Remarks to the Author):

Yumeng Zhang et al. described the diversity of tissue-derived mesenchymal stem cells (MSCs) and the presence of MSCs derived from endodermal tissue (DE-MSCs). hPSC-derived SOX17+ endodermal progenitors expressed typical MSC markers and had the ability to differentiate potential into the mesenchymal lineage. The scRNA-seq analysis revealed the appearance of DE-MSCs and culture conditions for MSC differentiation were examined. Intraperitoneal administration of DE-MSCs has therapeutic effects for ulcerative colitis model mice. This study demonstrates that DE-MSCs, which are endoderm-derived MSCs, have the potential to become a new cell source for regenerative medicine. The experimental concept and data analysis of this study are well established. However, there is a lack of data to support the author's claims. These comments have the potential to improve this paper for more better.

Major comments

- Figure 5e: The results of this experiment demonstrate that SOX17-positive cells derived from ES or iPS cells to the endoderm have the competency to give rise to mesenchymal-like cells. However, it remains questionable whether the results obtained under artificial in vitro induction conditions can be directly reflected in actual development in vivo. I believe that in vivo experiments using Sox17 reporter (or tracing) mice are essential to prove the model presented by the author (Model B) (Because the experiments using human tissues are not possible).

- Figure 6a: Intraperitoneal administration of human MSCs (hMSCs) improved symptoms in a mouse model of ulcerative colitis caused by DSS. What cytokines were elevated in the peripheral blood of the mice as a result of human cell transplantation? Where were the cells accumulated in the body? How long will these cells continue to produce cytokines in vivo?

- Figure 6e: According to the colon length data with mouse of ulcerative colitis model, Umbilical cord-MSCs seem to be more effective. Could you justify that DE-MSC is the suitable cell for ulcerative colitis? It is unclear the difference among HE staining tissue. Could you make a quantitative comparison among these samples? How much has the infiltration of immune cells changed? Are there any changes in the types of macrophages?

Minor comments

-The differentiation potential assay: Regarding the induction of mesenchymal lineage differentiation (bone, cartilage, fat) shown, it is unclear whether the MSCs are induced to mature cells or not. Gene expression specific to differentiated cells should be investigated and quantitatively displayed.

- Sup Figure 4d: Why is the expression of mesenchymal markers reduced after long-term Activin induction? In other words, please completely deny the possibility that contributions other than Sox17+ cells are responsible for MSC differentiation.

Reviewer #3 (Remarks to the Author):

Firstly, I would like to congratulate the authors of the manuscript for a detailed study which address a question relevant to the research field and presents novel findings. Regarding the statistical analysis of data, in my view the appropriate statistical tests were performed and appropriate conclusions were made by authors from the results obtained.

My recommendation would be to accept the manuscript for publication with major changes.

I have written my comments under each section to provide a clearer review.

Major questions and/or revision comments:

Organ-specific transcriptomes were observed in MSCs isolated from different embryonic Lineages.

- How were MSC isolated from colon, liver, adipose and umbilical cord? There is no protocol described in the 'methods' section or no references to protocols provided. How are the authors sure they have obtained MSC just by observing morphology? What other characteristics of these cells were tested?
- Authors have demonstrated that DE are effectively induced from H9 hESCs. One comment in regard to the differentiation of these cells to adipocytes, chondrocytes and osteocytes is that cell staining is not sufficient to demonstrate this. It would be beneficial to demonstrate the expression of at least one key factor which controls adipogenesis, chondrogenesis and osteogenesis. For example C/EBP β expression for adipogenesis, Sox9 or collagen 2 for chondrogenesis and runx2 for osteogenesis.

In addition to this von kossa stain which the authors used does stain for calcium deposits but is not necessarily associated with osteocytes. A more appropriate stain to detect osteocytes is alizarin red and/or immunohistochemistry for dentin matrix protein.

- Authors describe briefly how DE could also be induced from H1 hESCs and NL-1 hiPSCs however, the same rigorous tests authors performed on H9 cells was not described for these cells. Why not? In addition, could the authors describe the importance of the addition of results from H1 and NL-1 cells. Also, please describe what the difference is between H1 and H9 hESCs. If there are no differences in relation to this study why include the results from the H1 cells?

Signal modulations affected DE-MSK induction

- Authors need to provide a better explanation to the factors used and the combinations used. When was XAV-939 used ? Was it ever used in combination with CHIR and/or CHIR/SB? I have the understanding that XAV-939 is an inhibitor of wnt/B-catenin pathway, CHIR is a GSK3 inhibitor and SB an Activin/BMP/TFG inhibitor. Please explain how there is wnt activation. A better explanation within the introduction section and discussion would be beneficial to the manuscript.

DE-MSKs modulated inflammatory responses in cell culture and mouse model

- This section is an excellent addition to the manuscript and experiments are well designed. Results clearly demonstrate the author's observations. However, I was not able to find how pro and anti-inflammatory factors were measured. From figure 6 I presumed it was PCR through mRNA analysis/expression. Were there no cytokine ELISAS performed? These would support the PCR analysis.
- Figure 6e: panels have not error bars drawn on them.

Minor questions and/or revision comments:

- All primers used in PCR analysis should be listed in the 'methods' section along with primer sequences.
- More discussion on the in vivo transplant results (mouse model) should be discussed in the 'discussion'.
- Please correct a few spelling and formatting mistakes throughout the manuscript.

Reviewers' comments:

Reviewer #1 (Remarks to the Author):

The report by Zhang and collaborators is two-fold. One part describes the development *in vitro* of MSC (mesenchymal stem cell) like cells from human pluripotent cells experimentally committed in culture to endoderm development. This is an interesting, original observation, although more experiments are required before such an unexpected transition in culture can be validated (see detailed comments below). In parallel, the authors propose that these observations reflect the direct contribution, *in vivo*, of the endodermal germ layer to the emergence of a subset of specialized MSCs retaining expression of endodermal tissue markers. Data provided in support of this hypothesis are indirect and insufficient. Much more would be needed to make a convincing case.

Thanks for the encouraging comments about *in vitro* induction of DE-MSCs. We agree with your assessment that *in vitro* model only provides indirect evidence for *in vivo* processes. Based on your advice and editorial recommendation, we decided to focus on the *in vitro* model in this manuscript, and will pursue *in vivo* study in a separate project.

The manuscript is sometimes challenging to follow, due to the abundance of data presented (multiple conditions used in induction cultures, for instance, with no significant differences observed: see Fig4). The report would benefit from more concision and focus.

Thanks for the suggestion. In order to make the manuscript more concise, we removed the conditions that did not show significant differences and moved the original Fig. 4 to Supplementary Materials as Fig. S4.

Besides experimental results, a striking observation is that authors have ignored, in the introduction and discussion, directly relevant data produced by other groups. The fact that MSCs can be grown in culture from virtually all organs and tissues has been interpreted as reflecting their native association with blood vessels. Perivascular cells sorted to homogeneity – including from organs of partial endodermal origin like the pancreas and kidney – yield MSCs in culture, although the contribution of differently distributed progenitors cannot be formally excluded. This attributes a mesodermal origin to most MSCs, with the exception of pericytes from the cephalic region which originate in the neural crest and retain MSC potential *in vitro*. All these results, which have been the object of multiple publications, should have been discussed in the context of the present report.

Thanks for the suggestion. Perivascular cells related contents have been added in introduction and discussion.

Main other comments

- Introduction: “Mesenchymal stem/stromal cells (MSCs) are multipotent stem cells that widely exist in diverse tissues and organs in human body”. Discussion: “MSCs are integral components

of human body, and their heterogenous presence provides essential supports to various functions in different organs". Where is the evidence for these statements? While this is an interesting possibility, this reviewer is not aware of any published experiment that supports this contention. MSCs are, by definition, long-term cultured cells. Evidence that a functionally identical cell (multipotent, immunosuppressive...) naturally exists *in vivo* is still missing. All tissues contain cells which, upon culture, give rise to MSCs; that these innate cells be identical to their *in vitro* descent is not yet proven, albeit possible. The authors should be prudent, not readily extrapolate results obtained in culture, and moderate this statement.

Thanks for the informative comments. The statements in introduction and discussion have been modified to be more concise and moderate.

- For how many passages were MSCs from pre- and postnatal tissues maintained prior to experimental use (Fig.1)? This is an important information with respect to cellular heterogeneity, which could not be found.

Thanks for the comment. The cells were passaged five times before they were used for experiments in the manuscript. Relevant information has been added to Method section, under "Isolation and expansion of MSCs from specific human tissues".

- Regarding the point above, gene expression analysis on total MSC populations is difficult to interpret, with respect to the sensitivity of the techniques used and since these cultures are derived from total, unselected cell suspensions. Obviously, unrelated cells persist in these heterogeneous cell cultures for a while, notwithstanding the possibility of cell fusion in these conditions. The authors write "These data suggested that MSCs from different organs could retain tissue-specific expression". The fact that MSCs from different organs may have some distinct properties is actually supported by many other works, but it will take more than the data presented here to demonstrate that these differences reflect distinct germ line origins

Thanks for the suggestion. We agree that more *in vivo* study is necessary to determine the germ line origin of MSCs. Based on you and the editor's advice, we decided to focus on generation of DE-MSCs in cell culture, and removed the scRNA-seq analysis that implied *in vivo* origin of MSCs from endoderm organs. We removed the original Fig. 5e, and moved original Fig. 1b and c to Fig S1d and S1e.

The focus of this manuscript was changed to *in vitro* generation of DE-MSCs, not MSCs from different organs bearing distinct properties, and the data related to the later part was relocated from original Fig. 1 to supplemental Fig.1.

We also demonstrated that tissue specific MSCs generated for bulk RNA-seq are homogenous, and the percentages of MSC general marker CD105⁺ are shown below.

- The relevance to MSCs of data obtained by scRNAseq on “stromal” cells present in prenatal tissues and the adult rectum is very unclear... Please explain

The “stromal” identity was defined by the Enricher database which we used for cell type enrichment analysis (<https://maayanlab.cloud/Enrichr/>). Those cells are positive for CD73, CD90 and CD105. Based on the editorial recommendation, we decided not to discuss *in vivo* origin of endoderm MSCs, so we removed the section that analyzed scRNA-seq data of prenatal and adult rectum.

- The kidney and thymus are described as mesodermal; these are in fact endo-mesodermal organs. In addition, the mesoderm that contributes to thymus development is of ectodermal (neural crest) origin (so called mesectoderm), as precisely documented in animal chimaeras

Thanks for the comments. Because the scRNA-seq data of adult rectum has been removed in the revision based on editorial recommendation, the statements about kidney and thymus were deleted as well.

- It should be helpful to get a closer look at the morphology of cells in transition to MSCs (higher magnification, Fig.2b)

Thanks for the suggestion. We have increased the magnification to show the cell morphology. The images are shown in Fig. 1b.

Revised Fig. 1b

- Did the authors check whether their DE derived MSCs retain potential to differentiate into endodermal tissues?

Thanks for the great suggestion. We performed the experiments, and showed that DE-MSCs could be induced into cells expressing pancreatic markers *PDX1* and *PTF1A*. The results were added in Fig. S2c-d.

Fig. S2c-d

- Fig.2g: some stainings are sparse and higher magnifications of framed areas are not very informative. How long did cells cultured in differentiation media? Was the expression of genes activated in these three lineages monitored in parallel?

Trilineage differentiation experiments were conducted for 21 days. The information is provided in the Method section “Trilineage differentiation potential of DE-MSCs”. We also improved our experiments with optimized staining and additional assays. First, we replaced Von Kossa staining with Alizarin Red staining to improve the staining of osteocytes. Please see the results in the revised Figures (Fig. 1h, Fig. S1b, Fig. S2i, Fig. S4c, and Fig. S6c). Second, we used RT-qPCR to analyze adipogenic (*C/EBPβ*), chondrogenic (*COL2A1*) and osteogenic (*RUNX2*) markers. The results are shown in Fig. 1g, Fig. S1c, Fig. S4d and Fig. S6d. Below is an example of the results shown in Fig. 1g.

Fig. 1g

- The observed emergence of MSC like cells from Sox17+ cells is interesting and original.

However, not all DE cells exhibit bright Sox17 nuclear expression at d3 (supplem. Fig2). Also, the gate used to sort Sox17+ cells by FACS is not shown; did it include GFPlo cells? Did the authors verify that some early induced mesendoderm cells did not resist activin A induction, failing to differentiate into endoderm? Did they check the presence of cells expressing mesoderm markers (brachyury...) at successive times in culture? These are important controls, as the persistence of mesoderm cells might explain the emergence of MSCs in these cultures. The authors state indeed that “most hESCs became definitive endoderm progenitors”. The issue of the role of distinct signalings in early MSC induction has been examined in the paragraph “Signal modulations affected DE-MSC induction”, but in a rather confusing way. The authors are encouraged to describe these data more comprehensively

1. Thanks for the comments. The gate used to sort SOX17⁺ cells by FACS is now shown in Fig. S2f with undifferentiated hESCs as negative control. The SOX17⁻ and SOX17⁺ populations were well separated. To ensure that the DE-MSCs generated were not from SOX17⁻ contaminants, we only sorted out the top half of bright SOX17-GFP⁺ cells to demonstrate their ability to generate DE-MSCs.
2. We also checked the expression of mesoderm marker brachyury (*TBXT*) after DE induction. We showed that *TBXT* expression was not elevated during DE-MSC generation, when SOX17 mRNA decreased along with the emergence of DE-MSCs (see figure below). This suggested that DE-MSCs were not from the transdifferentiated cells. In a related project, a *TBXT* knockout cell line was established in our lab. We observed that *TBXT* K.O. hESCs could still differentiate to SOX17⁺ DE cells, and they were then induced into DE-MSCs as well.

RT-qPCR analysis for mRNA level of *TBXT* and *SOX17* on day 0, 3, 5, 10, 15 (Passage 0, P0), 18 (Passage 1, P1), 21 (Passage 2, P2), 24 (Passage 3, P3), and “meso” represents mesoderm progenitor generated by standard mesoderm induction (hESCs treated with Wnt pathway activator CHIR99021 for 24 hours)

- Fig. 5 describes MSCs derived from either “mesoderm”, a neural crest intermediate, or trophoblast, and illustrates expression by these cells of markers of diverse cell lineages representative of the respective germ lines of origin. However, there is no description at all of the methods and tissues used to derive these MSCs, culture conditions, number of passages, phenotype... as mentioned earlier, transcriptome results may be tainted by contaminating cells

from tissues of origin. No interpretation is possible with so little information. This part of the work should be thoroughly re-written

Thanks for the comments. The information has been added in the Method section under "Generation of MSCs with mesoderm, neural crest and trophoblast origin from hPSCs". Original Fig. 5 has been changed to Fig 3, and the gene expression analysis in Fig 3b-d was also discussed in the Result section.

We also showed that the cells for bulk RNA-seq were homogeneous MSCs with >91% CD44⁺ and > 93% CD105⁺. The data are shown in Fig. S6b.

- English spelling and style are of uneven quality; the main text is overall well written, while figure legends include obvious statements and unnecessary repeats but often lack important information

Thanks for the comments. The figure legends were edited.

Minor points

- Pages/lines should be numbered, which should make it much easier for reviewers to write comments

Pages numbers have been added.

- Paragraph titles in the Results section and Figure Legend should be written in present tense, not past: it gives the impression of one-time results that may not be reproduced...

Thanks for the advice. Paragraph titles in the Results and Figure Legends have been changed to present tense.

- CD44, CD73, and CD105 are not "MSC specific markers" (Results, paragraph 3)

Thank you for pointing this out. The statement has been changed.

Reviewer #2 (Remarks to the Author):

Yumeng Zhang et al. described the diversity of tissue-derived mesenchymal stem cells (MSCs) and the presence of MSCs derived from endodermal tissue (DE-MSCs). hPSC-derived SOX17⁺ endodermal progenitors expressed typical MSC markers and had the ability to differentiate potential into the mesenchymal lineage. The scRNA-seq analysis revealed the appearance of DE-MSCs and culture conditions for MSC differentiation were examined. Intraperitoneal administration of DE-MSCs has therapeutic effects for ulcerative colitis model mice. This study demonstrates that DE-MSCs, which are endoderm-derived MSCs, have the potential to become a new cell source for regenerative medicine. The experimental concept and data analysis of this

study are well established. However, there is a lack of data to support the author's claims. These comments have the potential to improve this paper for more better.

Thanks for the positive comments. Based on reviewers and editor's suggestions, we agree that the *in vitro* results were insufficient to support claims related to *in vivo* developmental processes, so we decided to focus on the *in vitro* study of DE-MSC induction in this manuscript. The *in vivo* study will be continued in a separate project.

Major comments

- Figure 5e: The results of this experiment demonstrate that SOX17-positive cells derived from ES or iPS cells to the endoderm have the competency to give rise to mesenchymal-like cells. However, it remains questionable whether the results obtained under artificial *in vitro* induction conditions can be directly reflected in actual development *in vivo*. I believe that *in vivo* experiments using Sox17 reporter (or tracing) mice are essential to prove the model presented by the author (Model B) (Because the experiments using human tissues are not possible).

Thank you for the comments. We took you and the editor's advice to focus on *in vitro* study in this manuscript. We removed the statement "endodermal organ MSCs partially originate from definitive endoderm *in vivo*", and Fig 5e was deleted as well. The *in vivo* study using mouse model will be conducted in a separate project.

- Figure 6a: Intraperitoneal administration of human MSCs (hMSCs) improved symptoms in a mouse model of ulcerative colitis caused by DSS. What cytokines were elevated in the peripheral blood of the mice as a result of human cell transplantation? Where were the cells accumulated in the body? How long will these cells continue to produce cytokines *in vivo*?

Thank you for your suggestion. ELISA assay was done to check what cytokine was elevated upon human cell transplantation. We investigated whether DE-MSCs (FBS) could upregulate anti-inflammatory cytokine TGF- β production in spleen cells. The treatment with DE-MSCs (FBS) significantly upregulated TGF- β level in spleen in comparison to the DSS+PBS control group (Fig. 4b).

Previous studies (PMID: 26864573 and 32384543) indicated that intraperitoneally administrated MSCs were no longer detected in the peritoneal lavage fluid after 20 minutes. The MSCs formed aggregates with mouse macrophages and lymphocytes, and attached to the walls of the peritoneal cavity. In addition, these MSCs usually disappeared in less than a week.

Fig. 4b

- Figure 6e: According to the colon length data with mouse of ulcerative colitis model, Umbilical cord-MSCs seem to be more effective. Could you justify that DE-MSC is the suitable cell for ulcerative colitis? It is unclear the difference among HE staining tissue. Could you make a quantitative comparison among these samples? How much has the infiltration of immune cells changed? Are there any changes in the types of macrophages?

Thanks for the suggestion. We demonstrated that DSS shortened colon length, and the symptom was significantly suppressed by DE-MSCs treatment. Similar beneficial effects were observed after the treatment with DE-MSCs and UC-MSCs (there is no significant difference between two of them) (Fig. 4d and e).

HE staining result was quantified to show the number of mice bearing obviously lesion in colon epithelium was reduced by treatment of DE-MSCs.

We demonstrated CD8⁺ T cell infiltration in DSS induced colitis mice (red signal in the image below), while DE-MSCs and UC-MSCs treatment significantly reduced CD8⁺ T cell infiltration (Fig. S7b).

Fig. S7b

Besides H&E staining, AB/PAS staining was also performed to detect the loss of mucin-producing goblet cells in DSS induced colitis mice. DE-MSCs and UC-MSCs treatment rescued the phenotype and ameliorated the pathological deterioration (Fig. S7c).

Fig. S7c

Minor comments

-The differentiation potential assay: Regarding the induction of mesenchymal lineage differentiation (bone, cartilage, fat) shown, it is unclear whether the MSCs are induced to mature cells or not. Gene expression specific to differentiated cells should be investigated and quantitatively displayed.

Thanks for the suggestion. We performed RT-qPCR analysis to detect adipogenic (*C/EBPβ*), chondrogenic (*COL2A1*) and osteogenic (*RUNX2*) markers. Please see the results in the revised Fig. 1g, Fig. S1c, Fig. S4d and Fig. S6d. Below is an example of results shown in Fig. 1g.

g

Fig. 1g

- Sup Figure 4d: Why is the expression of mesenchymal markers reduced after long-term Activin induction? In other words, please completely deny the possibility that contributions other than Sox17+ cells are responsible for MSC differentiation.

Our results suggested that long-term activin treatment probably drove hESCs to further differentiate to specific endoderm cell types, which led to the loss of plasticity to generate MSCs. In order to make the manuscript more concise, we removed the result of long-term Activin induction). In order to exclude the possibility that SOX17⁻ cells generated MSCs, we sorted out SOX17⁺ cells (Fig. S2f) and demonstrated that the purified SOX17⁺ cells gave rise to DE-MSCs.

Reviewer #3 (Remarks to the Author):

Firstly, I would like to congratulate the authors of the manuscript for a detailed study which address a question relevant to the research field and presents novel findings. Regarding the statistical analysis of data, in my view the appropriate statistical tests were performed and appropriate conclusions were made by authors from the results obtained.

My recommendation would be to accept the manuscript for publication with major changes.

Thanks for the positive comments.

I have written my comments under each section to provide a clearer review.

Major questions and/or revision comments:

Organ-specific transcriptomes were observed in MSCs isolated from different embryonic Lineages.

- How were MSC isolated from colon, liver, adipose and umbilical cord? There is no protocol described in the 'methods' section or no references to protocols provided. How are the authors sure they have obtained MSC just by observing morphology? What other characteristics of these cells were tested?

Thanks for the comments. The method of MSC isolation was added in the Method section under "Isolation and expansion of MSCs from specific human tissues". Isolated cells were analyzed for MSC markers, and we confirmed their trilineage differentiation capacity. The data are shown in Fig. S1b and S1c.

Fig. S1b and S1c

- Authors have demonstrated that DE are effectively induced from H9 hESCs. One comment in regard to the differentiation of these cells to adipocytes, chondrocytes and osteocytes is that cell staining is not sufficient to demonstrate this. It would be beneficial to, demonstrate the expression of at least one key factor which controls adipogenesis chondrogenesis and osteogenesis. For example *C/EBPβ* expression for adipogenesis, *Sox9* or collagen 2 for chondrogenesis and *runx2* for osteogenesis.

Thanks for the suggestion. RT-qPCR analysis was conducted to detect adipogenic (*C/EBPβ*), chondrogenic (*COL2A1*) and osteogenic (*RUNX2*) markers. Please see the data in Fig. 1g, Fig.S 1c, Fig. S4d, Fig. S6d. Below is an example of results shown in Fig. 1g.

g

Fig. 1g

In addition to this von kossa stain which the authors used does stain for calcium deposits but is not necessarily associated with osteocytes. A more appropriate stain to detect osteocytes is alizarin red and/or immunohistochemistry for dentin matrix protein.

Thanks for the suggestion. Alizarin Red was used to stain osteocytes in the revised manuscript. Please see the results in Fig. 1h, Fig. S1b, Fig. S2i, Fig. S4c and Fig. S6c. Below is an example of results shown in Fig. 1h.

h

Fig. 1h

- Authors describe briefly how DE could also be induced from H1 hESCs and NL-1 hiPSCs however, the same rigorous tests authors performed on H9 cells was not described for these cells. Why not? In addition, could the authors describe the importance of the addition of results from H1 and NL-1 cells. Also, please describe what the difference is between H1 and H9 hESCs. If there are no differences in relation to this study why include the results from the H1 cells?

Thanks for the comments. The same tests were conducted in H1 and NL-1 hPSCs, including cell surface marker expression (Fig. S2k) and trilineage differentiation (Fig. S2i). H1 is a male hESC

line, and H9 is a female hESC line. It is a standard practice in hPSC research to examine a differentiation method on multiple hESC and hiPSC lines to show the method is robust. We demonstrated that DE-MSCs could be effectively induced from both hESC and hiPSC lines with this method.

Fig. S2i

Signal modulations affected DE-MSC induction

- Authors need to provide a better explanation to the factors used and the combinations used. When was XAV-939 used? Was it ever used in combination with CHIR and/or CHIR/SB? I have the understanding that XAV-939 is an inhibitor of wnt/B-catenin pathway, CHIR is a GSK3 inhibitor and SB an Activin/BMP/TFG inhibitor. Please explain how there is wnt activation. A better explanation within the introduction section and discussion would be beneficial to the manuscript.

Thank you for the comments. CHIR99021, XAV939, SB431542 and their combination are applied from day 3 to day 5, for example, XAV939/ SB431542 in supplemental Fig. 4a means cells are treated with XAV939/ SB431542 under FBS condition or serum-free condition from day 3 to day 5.

We also added a brief description of small chemicals used in the manuscript. Small molecular signaling regulators was used from day 3 to day 5. CHIR is a GSK3 inhibitor that stabilizes β -catenin to activate WNT pathway.

DE-MSCs modulated inflammatory responses in cell culture and mouse model

- This section is an excellent addition to the manuscript and experiments are well designed. Results clearly demonstrate the author's observations. However, I was not able to find how pro and anti-inflammatory factors were measured. From figure 6 I presumed it was PCR through mRNA analysis/expression. Were there no cytokine ELISAS performed? These would support the PCR analysis.

Thanks for the comments and suggestion. In order to demonstrate the immunoregulatory ability of DE-MSCs, we exposed cells to proinflammatory factor IFN- γ , and then evaluated mRNA levels of proinflammatory cytokines (*IL-6*, *IL-8* and *CCL2*) and anti-inflammatory cytokines (*IDO1*, *PD-L1* and *TSG6*). RT-qPCR showed that IFN- γ upregulated the expression of *IL-6*, *IL-8* and *CCL2* in DE-MSCs and UC-MSCs. In respect to the anti-inflammatory genes, IFN- γ stimulation dramatically increased the mRNA levels of *IDO1* and *PD-L1* in both DE-MSCs and UC-MSCs (Fig. 4a and Fig. S7a).

We also investigated whether DE-MSCs (FBS) could upregulate anti-inflammatory cytokine TGF- β production in the supernatant of spleen cells by ELISA. We showed that DE-MSCs (FBS) significantly upregulated TGF- β level in the spleen compared to the DSS+PBS control group (Fig. 4b).

Fig. 4a

Fig. S7a

Fig. 4b

- Figure 6e: panels have not error bars drawn on them.

Original Figure 6 was changed to Figure 4, and error bars were added to all figures.

Minor questions and/or revision comments:

- All primers used in PCR analysis should be listed in the 'methods' section along with primer sequences.

Primer sequences are provided in this revised manuscript.

- More discussion on the *in vivo* transplant results (mouse model) should be discussed in the 'discussion'.

We have added discussions on the *In vivo* transplant results in Discussion section.

- Please correct a few spelling and formatting mistakes throughout the manuscript.

Thank you for the suggestion. Spelling and formatting mistakes have been corrected in this revised version.

Reviewers' comments:

Reviewer #1 (Remarks to the Author):

Zhang and collaborators have substantially revised their manuscript. The report is now much more focused. They now make a good case that endoderm lineage cells derived from pluripotent cells can produce MSCs in culture. Conversely, there is still no demonstration that in situ, endoderm lineage cells can convert to MSCs. Although the authors do not affirm this is the case, this remains as an underlying hypothesis. P 9: "Further analysis showed that each MSC line expressed a set of genes specific to its organ of origin. For example, colon-MSCs expressed intestine-associated genes, while liver-MSCs expressed liver-associated genes (Supplementary Fig. 1e)". This is strictly no proof that endoderm lineage cells in the liver and gut can contribute MSCs, since the analysis was performed on total, unseparated cultured cells from these organs. Although MSC "lines" are mentioned, these are not clonally derived. Only rigorous selection of endodermal lineage cells from these organs, followed by culture, might document an endodermal origin for MSCs in some organs. Albeit briefly mentioned by the authors, this point should be made clearer: MSCs can be derived from endo-mesodermal organs, where they have been shown to stem from the perivasculature, like in other organs. Whether endodermal cells per se can also yield MSCs in culture remains to be demonstrated.

The authors partly missed the point regarding the affiliation between native perivascular cells and culture derived mesenchymal stem cells, which should have been better discussed in the revised version (see above).

Independently of pluripotent cell systems, at least some MSCs from tissues located in the cephalic region are of neurectodermal (neural crest, via pericytes), and not mesodermal origin. This is another important illustration of diverse germ line origins for mesenchymal stem cells and must be mentioned in the manuscript.

Introduction: "blood vessel walls may be a source for MSCs in various organs". May be? There are tens and tens of publications establishing this fact...

Discussion: "MSCs are important part of human organs". This vague statement is not appropriate. As reminded in the original critique, and insufficiently discussed in the revised article, MSCs are by essence CULTURED cells. Whether identical cells exist naturally in vivo is unclear. It should be more appropriate to state that presumptive MSCs exist in organs (not only in human organs), the exact role of which in tissue turnover and repair is not known.

The authors should explain how their finding can be medically relevant ("DE-MSCs provide another potential tool to treat diseases in the near future").

English spelling should still be improved throughout.

Reviewer #2 (Remarks to the Author):

All the points raised in the first version of the manuscript have been properly addressed by the authors. I think this is an important paper for a field that deserves publication in Communications Biology.

Reviewers' comments:

Reviewer #1 (Remarks to the Author):

Zhang and collaborators have substantially revised their manuscript. The report is now much more focused. They now make a good case that endoderm lineage cells derived from pluripotent cells can produce MSCs in culture. Conversely, there is still no demonstration that in situ, endoderm lineage cells can convert to MSCs. Although the authors do not affirm this is the case, this remains as an underlying hypothesis. P9: "Further analysis showed that each MSC line expressed a set of genes specific to its organ of origin. For example, colon-MSCs expressed intestine-associated genes, while liver-MSCs expressed liver-associated genes (Supplementary Fig. 1e)". This is strictly no proof that endoderm lineage cells in the liver and gut can contribute MSCs, since the analysis was performed on total, unseparated cultured cells from these organs. Although MSC "lines" are mentioned, these are not clonally derived. Only rigorous selection of endodermal lineage cells from these organs, followed by culture, might document an endodermal origin for MSCs in some organs. Albeit briefly mentioned by the authors, this point should be made clearer: MSCs can be derived from endo-mesodermal organs, where they have been shown to stem from the perivascular, like in other organs. Whether endodermal cells per se can also yield MSCs in culture remains to be demonstrated.

Thanks for the comments. We agree with your opinion that there is strictly no proof that endoderm lineage cells in the liver and gut can contribute MSCs, therefore, we deleted the Supplementary Fig. 1e and all underlying hypothesis. And we also added "Whether endodermal cells can also yield MSCs in culture remains to be demonstrated." in Discussion section. And we also changed the title of this manuscript from "Definitive Endodermal Cells Supply a Novel Source of Mesenchymal Stem/Stromal Cells" to "Definitive Endodermal Cells Supply a Novel *in vitro* Source of Mesenchymal Stem/Stromal Cells".

The authors partly missed the point regarding the affiliation between native perivascular cells and culture derived mesenchymal stem cells, which should have been better discussed in the revised version (see above).

Affiliation between native perivascular cells and culture derived mesenchymal stem cells has been discussed in discussion section, the sentence "perivascular cells, including pericytes and adventitial cells, have proved as native sources of MSCs in developing and adult human organs⁶⁸" was added in Discussion section.

Independently of pluripotent cell systems, at least some MSCs from tissues located in the cephalic region are of neuroectodermal (neural crest, via pericytes), and not mesodermal origin. This is another important illustration of diverse germ line origins for mesenchymal stem cells and must be mentioned in the manuscript.

We have added this evidence in Introduction section.

Introduction: “blood vessel walls may be a source for MSCs in various organs”. May be? There are tens and tens of publications establishing this fact...

We deleted this sentence to prevent introducing underlying hypothesis on MSC *in vivo* origin, and added “pericytes are a clear source for MSCs isolated from various organs” in Introduction section.

Discussion: “MSCs are important part of human organs”. This vague statement is not appropriate. As reminded in the original critique, and insufficiently discussed in the revised article, MSCs are by essence CULTURED cells. Whether identical cells exist naturally *in vivo* is unclear. It should be more appropriate to state that presumptive MSCs exist in organs (not only in human organs), the exact role of which in tissue turnover and repair is not known. The authors should explain how their finding can be medically relevant (“DE-MSCs provide another potential tool to treat diseases in the near future”).

According to the reviewer’s kind suggestions, we have modified the statement about *in vivo* counterpart of MSCs in the beginning of the Discussion and explained the medical relevance of DE-MSCs in the end of the Discussion.

English spelling should still be improved throughout.

The English spelling has been improved.

REVIEWERS' COMMENTS:

Reviewer #1 (Remarks to the Author):

I thank the authors for their careful consideration of my remarks and suggestions, and appropriate revision of their original manuscript.